# Interstitial Lung Disease Is Associated with Sleep Disorders in Rheumatoid Arthritis Patients

**Natalia Mena-Vázquez** [1,2,*,†] **, Rocío Redondo-Rodriguez** [1,2,3,†] **, Pablo Cabezudo-García** [1,2,4] **,
Aimara Garcia-Studer** [1,2,3] **, Fernando Ortiz-Márquez** [1,2,3] **, Paula Borregón-Garrido** [1] **, Manuel Martín-Valverde** [3] **,
Inmaculada Ureña-Garnica** [1,2] **, Sara Manrique-Arija** [1,2,3] **, Laura Cano-García** [1,2]
**and Antonio Fernández-Nebro** [1,2,4]

1   Instituto de Investigación Biomédica de Málaga (IBIMA)-Plataforma Bionand, 29010 Malaga, Spain;
    rocio.redondo.sspa@juntadeandalucia.es (R.R.-R.); pablo.cabezudo.sspa@juntadeandalucia.es (P.C.-G.);
    aimara.garcia.sspa@juntadeandalucia.es (A.G.-S.); ferortmar@uma.es (F.O.-M.);
    paula16397@gmail.com (P.B.-G.); inmaculada.ureña.sspa@juntadeandalucia.es (I.U.-G.);
    sara.manrique@uma.es (S.M.-A.); laura.cano.sspa@juntadeandalucia.es (L.C.-G.);
    afernandezn@uma.es (A.F.-N.)
2   UGC de Reumatología, Hospital Regional Universitario de Málaga, 29009 Malaga, Spain
3   Departamento de Medicina y Dermatología, Universidad de Málaga, 29010 Malaga, Spain;
    manuelmartinvalver@gmail.com
4   UGC Neurociencia, Hospital Regional Universitario de Málaga, 29009 Malaga, Spain
*   Correspondence: natalia.mena.sspa@juntadeandalucia.es
†   These authors contributed equally to this work.

**Abstract:** Objective: To evaluate sleep disorders and associated factors in patients with rheumatoid-arthritis-associated interstitial lung disease (RA-ILD). Methods: We performed an observational study of 35 patients with RA-ILD (cases) and 35 age- and sex-matched RA patients without ILD (controls). We evaluated sleep disorders (Oviedo Sleep Questionnaire), positive psychological factors (resilience using the Wagnild and Young Resilience Scale, emotional intelligence using the 24-item Trait Meta-Mood Scale), anxiety and depression (Hospital Anxiety and Depression Scale), quality of life (36-item short-form survey), and fatigue (Functional Assessment of Chronic Illness Therapy Questionnaire). Other variables studied included the Charlson Comorbidity Index (CCI) and RA activity according to the DAS28-ESR. Results: Compared to the controls, the cases were characterized by poorer sleep quality with a higher prevalence of insomnia (42% vs. 20%; $p = 0.039$), greater severity of insomnia ($p = 0.001$), and lower sleep satisfaction ($p = 0.033$). They also had poorer resilience and emotional recovery and more severe anxiety and depression. A diagnosis of ILD was the only factor independently associated with the three dimensions of sleep quality. The predictors of poorer sleep satisfaction in patients with RA-ILD were age ($\beta = -0.379$), DAS28-ESR ($\beta = -0.331$), and usual interstitial pneumonia pattern ($\beta = -0.438$). The predictors of insomnia were DAS28-ESR ($\beta = 0.294$), resilience ($\beta = -0.352$), and CCI ($\beta = 0.377$). Conclusions: RA-ILD is associated with significant sleep disorders. RA-ILD seems to be an independent risk factor for sleep alterations, with a greater impact on insomnia. Age, disease activity, and comorbidity also play a role in sleep disorders in patients with RA-ILD.

**Keywords:** interstitial lung disease; rheumatoid arthritis; sleep disorders; insomnia; sleep satisfaction

## 1. Introduction

Rheumatoid arthritis (RA) is a chronic inflammatory autoimmune disease affecting around 1% of the general population [1,2]. While the exact etiology of RA is unknown, genetic, hormonal, and environmental factors are involved. In broad terms, a series of processes are triggered, including modification of autochthonous peptides, altered recognition of these by T cells, and production of autoantibodies by B cells [3]. The prevalent extra-articular expression of rheumatoid arthritis (RA) is associated with lung complications,

impacting approximately 60% of RA patients over the course of the disease [4,5], leading to high morbidity and mortality associated with lung damage, greater risk of infection, severity of arthritis, and co-occurrence with other conditions [6].

Patients with RA also experience psychosocial comorbid conditions, including mood disorders, which impair health-related quality of life and lead to sleep disorders [7]. Recent studies indicate that patients with RA are less satisfied with their sleep quality and more often experience insomnia than the general population [8,9]. These abnormalities can have many adverse effects on health, including increased pain, depression, cognitive impairment, systemic inflammation, and compromise of physical function [10–12]. Similarly, some studies suggest that elderly RA patients have poorer sleep quality than age- and sex-matched controls [13]. Furthermore, poor sleep quality, insomnia, and depression are also common in patients with all-cause ILD and independently associated with symptoms of depression and somnolence [14]; therefore, co-occurrence of RA and ILD can be expected to aggravate these problems.

Sleep quality and its effect on health can be evaluated from various standpoints. Thus, sleep satisfaction concerns the subjective perception of sleep quality, duration, and satisfaction. Insomnia, on the other hand, refers to difficulty falling and staying asleep, which may lead to insufficient and low-quality sleep. Lastly, hypersomnia refers to excessive daytime sleepiness that may interfere with activities of daily living and quality of life. Persons with hypersomnia sleep more than necessary yet feel tired and sleepy during the day. Hypersomnia not due to a nighttime sleep disorder or altered circadian rhythm may be caused by a central nervous system condition such as narcolepsy or Parkinson's disease [15].

In much the same way that sleep and mood disorders can negatively affect human health and outcomes in RA, the evidence also suggests an association between positive psychological states and better health outcomes, including a lower risk of cardiovascular disease and greater resistance to infection [16]. Therefore, we should also pay attention to the role of positive psychological factors such as resilience and emotional recovery when evaluating the psychosocial impact of disease. Resilience refers to the process, capacity, or outcome of successful adaptation despite challenges or threatening circumstances and has been associated, alongside other positive psychological factors, with improved health-related quality of life [17]. However, the impact of positive psychosocial factors, such as resilience, self-esteem, satisfactory social relationships, and ability to cope with life's challenges, has received less attention in patients with rheumatic diseases [18,19].

While poor sleep quality is prevalent in patients with all-cause interstitial lung disease, it may be even more pronounced in patients with RA-ILD due to the influence of exacerbating factors such as systemic inflammation, comorbidities, treatment, or disability associated with RA. Therefore, the objectives of the present study were as follows: (1) to explore sleep disorders in patients with RA-ILD; and (2) to identify factors associated with sleep disorders. Our study will help to better understand the needs of the patient with RA-ILD in terms of sleep and to design specific interventions aimed at improving the quality of life of affected patients.

## 2. Results

### 2.1. General Characteristics of the Study Population

The study population comprised 70 patients: 35 with RA-ILD (cases) and 35 with RA but not ILD (controls). The main characteristics of both groups are shown in Table 1. There were more men than women, and the mean age was 68 years. The median (IQR) follow-up for RA was 139.8 (79.5–220.6) months. The most frequent comorbid conditions for all patients were arterial hypertension (45.7%) followed by dyslipidemia (35.7%) and active smoking (24.2%).

**Table 1.** Baseline characteristics of the study population.

| Variable | RA-ILD N = 35 | RA without ILD N = 35 | *p*-Value |
|---|---|---|---|
| Clinical characteristics | | | |
| Age, years, mean (SD) | 69.7 (9.3) | 66.6 (7.0) | 0.130 |
| Age >60 years, *n* (%) | 27 (77.1) | 29 (82.9) | 0.550 |
| Male sex; *n* (%) | 20 (57.1) | 20 (57.1) | 1.000 |
| Duration of RA, months, median (IQR) | 149.8 (93.3–245.5) | 133.7 (67.8–204.2) | 0.384 |
| Duration of ILD, months, mean (SD) | 66.1 (47.2) | - | - |
| RF+ (>10), *n* (%) | 33 (94.3) | 31 (88.6) | 0.393 |
| High RF (>60), *n* (%) | 24 (68.6) | 17 (48.6) | 0.089 |
| ACPA+ (>20), *n* (%) | 32 (91.4) | 31 (88.6) | 0.690 |
| Radiographic erosions, *n* (%) | 21 (60.0) | 19 (55.6) | 0.705 |
| Comorbid conditions | | | |
| Smoking history | | | 0.760 |
|    Nonsmokers, *n* (%) | 17 (48.6) | 18 (51.4) | |
|    Ex-smokers, *n* (%) | 10 (28.6) | 8 (22.9) | |
|    Active smokers, *n* (%) | 8 (22.9) | 9 (25.7) | |
| Dyslipidemia, *n* (%) | 13 (37.1) | 12 (53.3) | 0.873 |
| Sleep apnea, *n* (%) | 4 (11.4) | 3 (8.6) | 0.690 |
| Arterial hypertension, *n* (%) | 19 (54.3) | 13 (37.1) | 0.150 |
| Obesity (BMI > 30), *n* (%) | 6 (17.1) | 6 (17.1) | 1.000 |
| Diabetes mellitus, *n* (%) | 7 (20.0) | 6 (17.1) | 0.759 |
| CCI, median (IQR) | 2.0 (1.0–3.0) | 1.0 (1.0–2.0) | 0.099 |
| Age-CCI, median (IQR) | 4.0 (3.0–5.0) | 3.0 (3.0–5.0) | 0.042 |
| Treatment | | | |
|   csDMARDs, *n* (%) | 28 (80.0) | 33 (94.3) | 0.074 |
|     Methotrexate, *n* (%) | 19 (54.3) | 27 (77.1) | 0.040 |
|     Leflunomide, *n* (%) | 3 (8.6) | 6 (17.1) | 0.284 |
|     Sulfasalazine, *n* (%) | 2 (5.7) | 2 (5.7) | 1.000 |
|     Hydroxychloroquine, *n* (%) | 6 (17.1) | 0 (0.0) | 0.010 |
|     Mycophenolate, *n* (%) | 4 (11.4) | 0 (0.0) | 0.032 |
|   bDMARDs, *n* (%) | 22 (62.9) | 18 (51.4) | 0.334 |
|     Anti-TNF, *n* (%) | 4 (11.4) | 11 (31.4) | 0.041 |
|     Tocilizumab, *n* (%) | 3 (8.6) | 2 (5.7) | 0.643 |
|     Abatacept, *n* (%) | 13 (37.1) | 3 (8.6) | 0.004 |
|     Rituximab, *n* (%) | 2 (5.7) | 0 (0.0) | 0.151 |
|   JAK inhibitors, *n* (%) | 0 (0.0) | 2 (5.7) | 0.151 |
|   Glucocorticoids, *n* (%) | 22 (62.9) | 6 (17.1) | 0.001 |
| Pulmonary function tests | | | |
| FVC < 80%, *n* (%) | 28 (80.0) | 5 (14.3) | <0.001 |
|   FVC predicted (%), mean (SD) | 63.0 (17.1) | 83.4 (4.4) | <0.001 |
| FEV1 < 80%, *n* (%) | 23 (67.6) | 5 (14.3) | 0.001 |
|   FEV1 predicted (%), mean (SD) | 68.7 (15.9) | 84.0 (11.5) | <0.001 |
| DLCO <80%, *n* (%) | 29 (85.3) | 3 (8.6) | <0.001 |
|   DLCO-SB predicted (%), mean (SD) | 61.0 (15.2) | 85.9 (7.9) | <0.001 |
| HRCT pattern | | | |
|   UIP, *n* (%) | 29 (82.9) | 0 (0.0) | <0.001 |
|   NSIP, *n* (%) | 6 (17.1) | 0 (0.0) | <0.001 |

Abbreviations. RA: rheumatoid arthritis; ILD: interstitial lung disease; SD: standard deviation; RF: rheumatoid factor; ACPA: anti-citrullinated peptide antibodies; CCI: Charlson Comorbidity Index (CCI); Age-CCI: age-adjusted CCI; csDMARDs: conventional synthetic disease-modifying antirheumatic drugs; bDMARDs: biological disease-modifying antirheumatic drugs; FVC: forced vital capacity; FEV1: forced expiratory volume in 1 s; DLCO: diffusing capacity of the lungs for carbon monoxide; HRCT: high-resolution computed tomography; UIP: usual interstitial pneumonia; NSIP: nonspecific interstitial pneumonia.

The HRCT scan was normal for all the controls, none of which had a history of lung symptoms. As can be seen in Table 1, both groups were well balanced for most of the clinical characteristics of RA, although median comorbidity according to the age-CCI was greater in patients with RA-ILD (*p* = 0.042). Moreover, Table 1 shows that the controls were treated mainly with methotrexate and anti-TNF agents, whereas the cases more frequently received

glucocorticoids (*p* = 0.001), hydroxychloroquine (*p* = 0.010), mycophenolate (*p* = 0.032), and abatacept (*p* = 0.004).

As for pulmonary function, HRCT revealed involvement for all the cases and none of the controls. The most common radiologic pattern on HRCT in the cases was UIP (82.9%), followed by NSIP (17.1%). Abnormal findings in PFT for the cases included reduced DLCO in 29 (85.3%) and reduced FVC in 28 (80.0%) (Table 1).

### 2.2. Characteristic of Sleep, Psychological Factors, and Quality of Life

Table 2 shows the results for sleep quality, psychological factors, mood disorders, and quality of life in both cases and controls. The cases were characterized by greater involvement in all three sleep dimensions compared to the controls, with poorer sleep satisfaction (*p* = 0.033) and more severe insomnia (*p* = 0.001) and hypersomnia (*p* = 0.005). The prevalence of insomnia in patients with RA-ILD according to the criteria of the ICD-10 was 42.9%. The prevalence of insomnia was greater in cases than in controls (OR (95% CI): 3.000 (1.034–8.702); *p* = 0.039). The prevalence of hypersomnia in cases according to ICD-10 was 5.7% (Table 2).

**Table 2.** Sleep and psychosocial disorders in patients with RA with and without ILD.

| Variable | RA-ILD N = 35 | RA without ILD N = 35 | *p*-Value |
|---|---|---|---|
| Sleep quality (OSQ) | | | |
| Subjective satisfaction with sleep, mean (SD) | 3.5 (0.9) | 4.0 (0.8) | 0.033 |
| Severity of insomnia, mean (SD) | 17.9 (7.8) | 11.7 (4.8) | 0.001 |
| Insomnia ICD-10, *n* (%) | 15 (42.9) | 7 (20.0) | 0.039 |
| Hypersomnia, mean (SD) | 5.8 (2.3) | 4.4 (1.4) | 0.005 |
| Hypersomnia ICD-10 | 2 (5.7) | 1 (2.9) | 0.555 |
| Positive psychological factors | | | |
| Resilience, mean (SD) | 97.1 (30.6) | 111.7 (22.8) | 0.027 |
| Emotional intelligence (TMMS-24) | | | |
| Emotional attention, mean (SD) | 22.1 (6.8) | 23.8 (5.6) | 0.248 |
| Low, *n* (%) | 13 (37.1) | 9 (25.7) | 0.584 |
| Adequate, *n* (%) | 21 (60.0) | 25 (71.4) | 0.584 |
| High, *n* (%) | 1 (2.9) | 1 (2.9) | 0.584 |
| Emotional clarity, mean (SD) | 22.1 (7.5) | 24.6 (4.6) | 0.089 |
| Low, *n* (%) | 16 (45.7) | 6 (17.1) | 0.023 |
| Adequate, *n* (%) | 17 (48.6) | 28 (80.0) | 0.023 |
| High, *n* (%) | 2 (5.7) | 1 (2.9) | 0.023 |
| Emotional recovery, mean (SD) | 20.2 (8.3) | 25.3 (6.4) | 0.006 |
| Low, *n* (%) | 17 (48.6) | 3 (8.6) | 0.002 |
| Adequate, *n* (%) | 16 (45.7) | 28 (80.0) | 0.002 |
| High, *n* (%) | 2 (5.7) | 4 (11.4) | 0.002 |
| Mood disorders (HADS scale) | | | |
| Anxiety, mean (SD) | 7.0 (4.0–9.0) | 3.0 (0.0–5.0) | 0.006 |
| Anxiety ≥11, *n* (%) | 8 (22.9) | 3 (8.6) | 0.101 |
| Depression, median (IQR) | 6.5 (3.0–9.5) | 3.0 (2.0–6.0) | 0.008 |
| Depression ≥11, *n* (%) | 10 (28.6) | 5 (14.3) | 0.145 |
| Inflammatory activity | | | |
| DAS28-ESR, mean (SD) | 3.2 (1.1) | 2.8 (1.0) | 0.028 |
| Remission-low activity, *n* (%) | 66 (60.0) | 68 (66.7) | 0.315 |
| Moderate-high activity, *n* (%) | 44 (40.0) | 34 (33.3) | 0.315 |
| VAS pain, median (IQR) | 4.0 (3.0–6.0) | 4.0 (2.0–5.0) | 0.283 |
| VAS general patient, median (IQR) | 5.0 (3.0–6.0) | 3.0 (1.0–4.0) | 0.046 |
| Physical function | | | |
| HAQ, median (IQR) | 1.2 (0.6) | 0.7 (0.6) | 0.003 |

**Table 2.** *Cont.*

| Variable | RA-ILD N = 35 | RA without ILD N = 35 | *p*-Value |
|---|---|---|---|
| Fatigue | | | |
| FACIT, median (IQR) | 21.6 (7.1) | 20.4 (6.6) | 0.461 |
| Quality of life | | | |
| SF-36 PCS, mean (SD) | 30.4 (9.8) | 35.7 (10.9) | 0.032 |
| SF-36 MCS, median (IQR) | 37.6 (11.7) | 44.2 (12.8) | 0.038 |

Abbreviations: RA: rheumatoid arthritis; ILD: interstitial lung disease; SD: standard deviation; RF: rheumatoid factor; IQR: interquartile range; DAS28-ESR: 28-joint Disease Activity Score with erythrocyte sedimentation rate; 36-item Short Form Survey (SF-36) PCS: physical component summary; SF-36 MCS: mental component summary; HAQ: Health Assessment Questionnaire; VAS: visual analog scale; FACIT: Functional Assessment of Chronic Illness Therapy—Fatigue scale.

Furthermore, in comparison with the controls, the cases had lower values for positive psychological factors, such as resilience (*p* = 0.027) and emotional recovery (*p* = 0.006), while also presenting higher levels of anxiety (*p* = 0.006) and depression (*p* = 0.008). Additionally, cases had poorer results on the quality of life questionnaire (SF-36 physical component summary (PCS), *p* = 0.032; SF-36 mental component summary (MCS), *p* = 0.038) and poorer physical function according to the HAQ (*p* = 0.003) (Table 2).

While disease activity remained low for all the participants, DAS28-ESR values were higher in the cases than in the controls (mean (SD), 3.2 (1.2) vs. 2.8 (1.0); *p* = 0.028).

*2.3. Correlations between Sleep and Clinical and Psychological Characteristics in Cases and Controls*

Supplementary Table S1 shows results for the correlation between subjective sleep satisfaction, insomnia, hypersomnia, and various clinical, psychological, and functional characteristics in the whole sample. As can be seen, a significant negative correlation was found with age, inflammatory activity, and mood disorders, indicating that these factors can negatively affect sleep quality in patients with RA. In contrast, a significant positive correlation was found between sleep satisfaction and emotional recovery. Negative correlations were recorded for insomnia and hypersomnia and positive psychological factors such as resilience, emotional attention, clarity, and recovery.

Figure 1 shows a heat map of the correlation coefficients for sleep disorders in patients with RA-ILD and clinical variables, pulmonary function, and psychological variables. Sleep satisfaction and insomnia were the sleep disorders that showed the greatest number of correlations with the characteristics of patients with RA-ILD. Both conditions were associated with the duration of ILD, inflammatory activity, mood disorders, and FVC. Similarly, insomnia was negatively associated with resilience, whereas sleep satisfaction and hypersomnia were negatively associated with age. Both insomnia and hypersomnia were positively associated with the comorbid conditions included in the age-CCI.

*2.4. Multivariate Analysis*

An initial analysis based on the whole sample was performed to evaluate the weight of ILD in sleep quality. Table 3 shows the results of the univariate and multivariate analyses of factors associated with sleep quality in all patients with RA. The dependent variables for the 3 models were sleep satisfaction, insomnia, and hypersomnia. As possible predictive factors, we evaluated sex, age, diagnosis of ILD, DAS28-ESR, resilience, emotional recovery, depression, and age-CCI. As can be seen in the multivariate analysis, after adjusting for the other variables, a diagnosis of ILD was the only factor that maintained its association with the three dimensions of sleep quality. Sleep satisfaction was associated with age >60 years (β = −0.318), diagnosis of ILD (β = −0.222), and DAS28-ESR values (β = −0.254), which maintained a significant negative association with sleep satisfaction. In the case of insomnia, the variables that remained significant after adjustment were ILD (β = 0.280) and resilience (β = −0.423). Finally, hypersomnia was associated with a diagnosis of ILD

(β = 0.242) and emotional recovery (β = −0.303). We conducted an alternative multivariate model, adjusting for treatment with methotrexate, biological therapy, mycophenolate, and glucocorticoids, with comparable results (Supplementary Table S2).

**Figure 1.** Heatmap showing correlations between sleep quality measured using the Oviedo Sleep Questionnaire and psychosocial and epidemiological factors and factors associated with RA-ILD.

**Table 3.** Univariate and multivariate analyses of the characteristics associated with sleep disorders in RA.

| Predictor | Univariate | | Multivariate | | |
|---|---|---|---|---|---|
| | B | 95% CI | B | 95% CI | *p*-Value |
| Sleep satisfaction * | | | | | |
| Female sex | 0.308 | −0.134, 0.751 | | | |
| Age > 60 years | −0.771 | −1.310, −0.231 | −0.730 | −1.223, −0.237 | 0.004 |
| ILD | −0.486 | −0.914, −0.058 | −0.408 | −0.814, −0.020 | 0.048 |
| Age-CCI | −0.162 | −0.288, −0.035 | | | |
| DAS28-ESR | −0.326 | −0.544, −0.109 | −0.242 | −0.434, −0.030 | 0.026 |
| Resilience | 0.007 | −0.001, 0.015 | | | |
| ER | 0.034 | 0.007, 0.062 | | | |
| Depression | −0.077 | −0.127, −0.027 | | | |
| Severity of insomnia ** | | | | | |
| Female sex | 0.392 | −3.101, 3.884 | | | |
| Age > 60 years | −0.099 | −4.544, 4.347 | | | |
| ILD | 6.257 | 3.148, 9.366 | 3.995 | 1.125, 6.865 | 0.007 |
| Age-CCI | 1.240 | 0.291, 2.430 | | | |
| DAS28-ESR | 2.494 | 0.797, 4.192 | | | |
| Resilience | −0.130 | −0.184, −0.076 | −0.109 | −0.160, −0.058 | <0.001 |
| ER | −0.138 | −0.357, 0.081 | | | |
| Depression | 0.721 | 0.343, 1.099 | | | |

**Table 3.** *Cont.*

| Predictor | Univariate | | Multivariate | | |
|---|---|---|---|---|---|
| | **B** | **95% CI** | **B** | **95% CI** | ***p*-Value** |
| Hypersomnia *** | | | | | |
| Female sex | 0.417 | −0.584, 1.417 | | | |
| Age > 60 years | 1.320 | 0.332, 2.308 | | | |
| ILD | 1.400 | 0.465, 2.335 | 0.995 | 0.045, 1.945 | 0.040 |
| Age-CCI | 0.329 | 0.043, 0.615 | | | |
| DAS28-ESR | 0.139 | −0.379, 0.656 | | | |
| Resilience | −0.020 | −0.037, −0.003 | | | |
| ER | −0.100 | −0.159, −0.042 | −0.080 | −0.141, −0.029 | 0.011 |
| Depression | 0.103 | −0.014, 0.220 | | | |

* Nagelkerke R2 = 0.216; ** Nagelkerke R2 = 0.382; *** Nagelkerke R2 = 0.174. Variables included in the equation: sex, age, diagnosis of ILD, DAS28-ESR, resilience, ER, depression, age-CCI. Abbreviations. RA: rheumatoid arthritis; ILD: interstitial lung disease; Age-CCI: age-adjusted Charlson Comorbidity Index; DAS28-ESR: 28-joint Disease Activity Score with erythrocyte sedimentation rate; ER: emotional recovery.

We subsequently ran a further three multivariate linear regression models to identify the main factors associated with each dimension of sleep quality (DV: sleep satisfaction, insomnia, and hypersomnia) only in patients with RA-ILD (Table 4). As can be seen, subjective sleep satisfaction in RA-ILD was independently associated with age ($\beta = -0.379$), inflammatory activity as measured by DAS28-ESR ($\beta = -0.331$), and UIP radiologic patterns ($\beta = -0.438$). Insomnia was associated with inflammatory activity as measured by DAS28-ESR ($\beta = 0.294$), resilience ($\beta = -0.352$), and age-CCI ($\beta = 0.377$), whereas hypersomnia was only associated with age ($\beta = 0.401$).

**Table 4.** Univariate and multivariate analyses of characteristics associated with sleep disorders in RA-ILD.

| Predictor | Univariate | | Multivariate | | |
|---|---|---|---|---|---|
| | **B** | **95% CI** | **B** | **95% CI** | ***p*-Value** |
| Sleep satisfaction * | | | | | |
| Female sex | 0.150 | −0.496, 0.796 | | | |
| Age > 60 years | −0.792 | −1.503, −0.081 | −0.817 | −1.458, −0.177 | 0.014 |
| Age-CCI | −0.054 | −0.228, 0.120 | | | |
| FVC | 0.019 | 0.001, 0.037 | | | |
| UIP pattern | −1.292 | −0.240, −2.343 | −1.419 | −2.344, −0.493 | 0.004 |
| DAS28-ESR | −0.334 | −0.659, −0.009 | −0.324 | −0.601, −0.046 | 0.024 |
| Resilience | 0.008 | −0.002, 0.018 | | | |
| Depression | −0.057 | −0.133, 0.020 | | | |
| Severity of insomnia ** | | | | | |
| Female sex | −1.583 | −7.065, 3.898 | | | |
| Age > 60 years | 0.449 | −6.043, 6.491 | | | |
| Age-CCI | 1.869 | 0.536, 3.202 | 1.584 | 0.499, 2.669 | 0.006 |
| FVC | −0.113 | −0.272, 0.046 | | | |
| UIP pattern | 3.979 | −5.659, 10.618 | | | |
| DAS28-ESR | 3.423 | 0.743, 6.103 | 2.436 | 1.321–5.194 | 0.041 |
| Resilience | −0.118 | −0.197, −0.038 | −0.114 | −0.179, −0.048 | 0.001 |
| Depression | 0.608 | −0.029, 1.244 | | | |

**Table 4.** *Cont.*

| Predictor | Univariate | | Multivariate | | |
|---|---|---|---|---|---|
| | B | 95% CI | B | 95% CI | *p*-Value |
| Hypersomnia *** | | | | | |
| Female sex | 0.650 | −0.988, 2.288 | | | |
| Age > 60 years | 2.013 | 0.281, 3.745 | 2.208 | 0.423–3.994 | 0.017 |
| Age-CCI | 0.452 | 0.035, 0.869 | | | |
| FVC | 0.007 | −0.042, 0.057 | | | |
| UIP pattern | 2.000 | −0.837, 4.837 | | | |
| DAS28-ESR | −0.342 | −1.216, 0.533 | | | |
| Resilience | −0.004 | −0.031, 0.023 | | | |
| Depression | −0.038 | −0.234, 0.169 | | | |

* Nagelkerke R2 = 0.353; ** Nagelkerke R2 = 0.464; *** Nagelkerke R2 = 0.136. Variables included in the equation: sex, age, age-CCI, UIP radiologic pattern, DAS28-ESR, resilience, depression. Abbreviations. RA: rheumatoid arthritis; ILD: interstitial lung disease; Age-CCI: age-adjusted Charlson Comorbidity Index; FVC: forced vital capacity; UIP: usual interstitial pneumonia; HAQ: Health Assessment Questionnaire; DAS28-ESR: 28-joint Disease Activity Score with erythrocyte sedimentation rate.

## 3. Discussion

Sleep disorders such as, insomnia and hypersomnia, are common. They affect 5–40% of the general population and constitute a public health problem [20]. Their prevalence is increasing [21]. These variations are due mainly to differences in the methodology used to define terms and collect data. In the present study, we evaluated sleep disorders using a semistructured interview based on the OSQ and were able to define the prevalence of insomnia and hypersomnia in line with the criteria of the ICD-10. Some sleep questionnaires address a single dimension. For example, the Pittsburgh Sleep Quality Index or the Medical Outcomes Study Sleep Scale (MOS-Sleep), which evaluate the subjective quality of sleep [22,23]. The OSQ, on the other hand, evaluates sleep satisfaction and facilitates the diagnosis of insomnia and hypersomnia according to the diagnostic criteria of ICD-10 and DSM-IV. The OSQ has been used in patients with rheumatic disease [19,24] and other disorders [25,26]. Furthermore, the ICD-10 correlates well with the third edition of the International Classification of Sleep Disorders (ICSD-3) and DSM-V [15].

In accordance with these methods, findings for patients with RA-ILD were more severe in all sleep dimensions (satisfaction, insomnia, and hypersomnia), and insomnia was more prevalent in RA-ILD patients than in patients with RA but not ILD. A diagnosis of ILD was the only factor that maintained its association with the three sleep dimensions in the multivariate models. While no studies specifically address sleep disorders in RA-ILD, some authors report that patients with RA or idiopathic pulmonary fibrosis experience greater impairment of sleep quality [8,9,27,28]. In fact, recent studies suggest a high prevalence of sleep problems in patients with pulmonary fibrosis; this seems to negatively affect quality of life, disease progression, and survival [27,28]. Therefore, we might expect co-occurrence of RA and ILD to increase the likelihood of sleep disorders. In fact, the prevalence of insomnia in cases was higher than in controls. The elevated prevalence of comorbid conditions heightened joint disease severity as indicated by DAS28-ESR, inflammation, and increased resilience impairment observed in patients with RA-ILD suggest potential links to the pathogenic mechanisms of insomnia.

Sleep satisfaction, both in controls and in cases, was associated with age and with inflammatory activity according to DAS28-ESR. Similar observations have been made in other studies examining inflammation in the general population [29–31] and in RA [8,10]. A recent meta-analysis found that patients with worse sleep quality had higher levels of inflammation according to their C-reactive protein and IL-6 values [29]. Similarly, Katz et al. [8] found inflammatory activity and pain to be significantly associated with poor sleep quality in patients with RA. Moreover, both the architecture and depth of sleep are affected with age [32]. Older persons have been reported to spend more time in light sleep stages than in deep sleep stages, leading them to wake more frequently (sleep

fragmentation) [33,34]. Furthermore, the melatonin peak decreases markedly at night in older persons [35], potentially leading to variation in the circadian rhythm not only in terms of sleep but also in terms of other bodily functions. Thus, aging is usually accompanied by advanced sleep phase syndrome since older persons tend to go to bed and wake earlier than younger adults. This syndrome can also worsen daytime hypersomnia [36,37].

Poor sleep quality is associated with greater physical and mental morbidity and entails high social and health costs in terms of accidents, absenteeism, and medical expenses. Patients with RA, especially those with RA-ILD, more often experience comorbid conditions, including psychosocial factors, than controls [6]. Sleep problems, which affect quality and duration, have been associated with greater morbidity, especially in older adults, in terms of a greater risk of falls, cognitive impairment, and all-cause mortality [38]. According to these results, patients with RA-ILD in the present study more frequently had comorbid conditions than the controls. In addition, an association was observed between the CCI and insomnia.

We also found the UIP radiologic pattern to be associated with lower sleep satisfaction in the cases. Other authors report that the most fibrotic forms of ILD, especially UIP-type ILD, have been associated with poorer quality of life, greater disease progression, and mortality in affected patients [39]. Moreover, this type of interstitial involvement tends to respond worse to treatment, and the course of the disease is similar to that of idiopathic pulmonary fibrosis [40].

Furthermore, insomnia was inversely associated with resilience in both cases and controls. We were unable to find another study that directly correlated sleep disorders with positive psychological factors, such as resilience, in patients with RA and patients with RA-ILD. However, this correlation has been reported in patients with other chronic inflammatory joint diseases, such as spondyloarthritis and psoriatic arthritis [19]. Similarly, low scores for resilience and emotional recovery in the general population have been shown to contribute to the development and maintenance of sleep disorders such as insomnia and hypersomnia [41], probably because patients with insomnia are less able to overcome stress by means of an adaptive response, in turn negatively affecting emotional regulation and favoring the onset and maintenance of insomnia [42]. This could also account for the association we observed between hypersomnia and poorer emotional recovery in patients with RA triggered by their being less able to adapt and cope emotionally as a result of their disease.

Our study is subject to a series of limitations. First, it is the only study to date to describe sleep disorders and their association with other psychosocial factors in patients with RA with and without ILD. However, its cross-sectional design prevents us from establishing causal relationships between the results recorded. Second, the correlations between sleep disorders and the remaining clinical and psychosocial variables were mainly weak or moderate. In this sense, it must be stressed that sleep quality is a very complex variable resulting from multiple personal, physical, and psychosocial factors, and not all of these could be included in the study. Nevertheless, the multivariate analysis revealed relevant factors associated with sleep disorders. In fact, one of the strengths of the study was the use of the OSQ to measure sleep quality. This instrument has been validated and enables an exhaustive assessment of the sleep–wake cycle in three main dimensions—namely, satisfaction, insomnia, and hypersomnia—which made it possible to identify several associated factors separately [25]. We must also take into account the role of factors that may impact sleep disorders, such as occupation types, diet, stress, and drug use. Nevertheless, despite these considerations, we were able to demonstrate disease characteristics, including the duration of ILD, inflammatory activity, and radiological pattern, which were associated with sleep disorders in patients with RA-ILD. On the other hand, our sample size might have been too small to detect more substantial differences; nevertheless, we did observe variances in the sleep quality between patients with RA-ILD and those with RA without ILD. Finally, while the OSQ was designed for patients with severe mental illness, it can also be used in other settings and diseases, including rheumatic disease [19].

## 4. Materials and Methods

### 4.1. Design and Data Source

We performed an observational case–control study based on a prospective cohort of patients with RA-ILD and compared them to a group of patients without RA-ILD. The study was carried out in the Rheumatology Department of the Hospital Regional Universitario de Málaga (HRUM) and approved by the Ethics Committee of HRUM (2627-N-21). All participants gave their written informed consent before entering the study.

#### 4.1.1. Cases

The case group comprised patients with RA-ILD selected consecutively between 2021 and 2022 from the RA-ILD cohort of HRUM, which has been prospectively followed up since 2015. The inclusion criteria were as follows: (1) age $\geq$ 16 years; (2) diagnosis of RA according to the criteria of the American College of Rheumatology/European League Against Rheumatism 2010 (ACR/EULAR 2010 criteria); and (3) ILD according to the criteria of the American Thoracic Society/European Respiratory. The exclusion criteria were as follows: (1) co-occurrence of an inflammatory disease other than RA (secondary Sjögren syndrome allowed); (2) pregnancy or breastfeeding; and (3) illiteracy or mental illness that hampered reading–writing and comprehension.

#### 4.1.2. Controls

The control group comprised patients with RA in whom ILD was ruled out. These patients were selected consecutively from the prospective RA cohort at HRUM. Controls were matched with cases by age, sex, and disease duration. The inclusion criteria for the controls were as follows: (1) age $\geq$ 16 years; (2) diagnosis of RA according to the classification criteria of the American College of Rheumatology/European League Against Rheumatism 2010 (ACR/EULAR 2010 criteria); (3) no respiratory conditions such as dyspnea or cough; and (4) normal findings in pulmonary function testing (PFT) and high-resolution computed tomography (HRCT) of the chest. The exclusion criteria for controls were the same as for cases.

### 4.2. Study Protocol

After signing the informed consent document, all participants who attended the clinic between 2021 and 2022 and who met the inclusion criteria were seen according to the pre-established data collection protocol. Participants underwent a blood analysis and completed the questionnaires. All participants underwent PFT and an HRCT scan at inclusion if recent data for these investigations were not available.

### 4.3. Outcome Measures and Definitions

#### 4.3.1. Variables: Sleep, Psychosocial Factors, and Physical Functioning

Sleep quality was evaluated using the Oviedo Sleep Questionnaire (OSQ) [25]. The questionnaire is a semistructured interview comprising 15 items used to diagnose insomnia and hypersomnia according to the diagnostic criteria of the International Classification of Diseases, Tenth Revision (ICD-10) and the Diagnostic and Statistical Manual of Mental Disorders, Fourth Edition (DSM-IV). The first 13 items are subdivided into 3 subscales that measure the following during the previous month: (1) subjective satisfaction with sleep (OSQ1); (2) the nature of insomnia, its effects on wakefulness, and its severity (9 items: OSQ2.1 to OSQ2.4, OSQ3 to OSQ7); and (3) hypersomnia (3 items: OSQ2.5, OSQ8, and OSQ9). For subjective satisfaction with sleep (scores ranging from 1 to 7), a higher score indicates greater satisfaction, whereas for insomnia and hypersomnia, the highest scores indicate greater severity (scores ranging from 9 to 45 and from 3 to 15, respectively). The 9 items make up the OSQ. There is no OSQ scale for severity of hypersomnia.

A patient was diagnosed with insomnia based on the criteria of the ICD-10 [43], that is, at least 1 of the 4 items (OSQ2.1 to OSQ2.4) is present $\geq$ 3 days per week (i.e., reaching a value of at least 3 or more) and item OSQ7 is present $\geq$ 3 days per week (i.e., scoring 3 or

more). A patient was diagnosed with hypersomnia when items OSQ2.1 to OSQ2.4 did not reveal difficulties with nighttime sleep (i.e., all scoring 1) and items OSQ2.5, OSQ8, and OSQ9 were present 6–7 days per week (i.e., scoring 5).

The positive psychosocial factors assessed were resilience and emotional intelligence. Resilience was evaluated using the Wagnild and Young Resilience Scale [44], which is scored from 25 to 175 points and assigns higher scores to greater levels of resilience. Emotional intelligence was measured using the 24-item Trait Meta-Mood Scale (TMMS-24) [45,46], which is used to evaluate 3 dimensions of emotional intelligence, namely, emotional attention, clarity, and repair. Respondents had to provide their degree of agreement or disagreement for each item, with 1 representing no agreement and 5 representing total agreement. Emotional attention, clarity, and repair were classified as low, adequate, or high according to the scores (Supplementary Table S3).

Negative psychosocial aspects included anxiety and depression, which were evaluated using the Hospital Anxiety and Depression Scale (HADS) [47,48]. HADS is scored from 0 to 7 points when there is no evident disease and between 8 and 10 in cases of doubt. A score $\geq 11$ in each of the subscales indicates a positive case. Pain was evaluated using the visual analog scale, ranging from 0 (no pain) to 10 (maximum pain), quality of life using the 36-item Short Form Survey (SF-36) [49], physical functioning using the Health Assessment Questionnaire (HAQ) [50], and fatigue using the Functional Assessment of Chronic Illness Therapy—Fatigue scale (FACIT) [51]. The score on the FACIT questionnaire ranges from 0 to 52 points, with higher values indicating lower degrees of fatigue.

### 4.3.2. Variables Associated with Interstitial Lung Disease

The presence of ILD and the different patterns were defined according to the standard criteria of the American Thoracic Society/European Respiratory Society International Multidisciplinary Consensus Classification of the Idiopathic Interstitial Pneumonias based on lung biopsy or HRCT. The 3 specific patterns are nonspecific interstitial pneumonia (NSIP), usual interstitial pneumonia (UIP), and other patterns, such as bronchiolitis obliterans, organizing pneumonia, lymphoid interstitial pneumonia, and mixed patterns [52]. PFT included complete spirometry, expressed as percent predicted and adjusted for age, sex, and height. Forced vital capacity (FVC) was considered abnormal if <80% predicted. Diffusing capacity of the lung for carbon monoxide was evaluated using the single-breath method (DLCO-SB) and considered abnormal if <80% predicted [53].

### 4.3.3. Other Variables

The variables associated with RA included time with symptoms and diagnostic delay. Smoking history (active, former, and never) was updated at inclusion, as were risky cardiovascular comorbid conditions (e.g., arterial hypertension, diabetes mellitus, dyslipidemia, and obesity) and general comorbid conditions included in the Charlson Comorbidity Index (CCI) and age-adjusted CCI (age-CCI (adjustment by age groups)) [54,55]. The diagnosis of sleep apnea was also included. Joint inflammatory activity was evaluated using the 28-joint Disease Activity Score with erythrocyte sedimentation rate (DAS28-ESR), as follows: moderate–high, $\geq 3.2$; low–remission, <3.2). We also collected information on treatment at the cutoff and before as follows: conventional synthetic disease-modifying antirheumatic drugs (csDMARDs), biologic DMARDs (bDMARDs), other immunosuppressants, and glucocorticoids.

### 4.4. Statistical Analysis

We first performed a descriptive analysis of the main variables. Qualitative variables were expressed as absolute frequencies and percentages; quantitative variables were expressed as mean (standard deviation (SD)) or median (interquartile range (IQR)). Normality was verified using the Kolmogorov–Smirnov test.

Clinical characteristics, sleep disorders, and other psychosocial factors were compared between patients with RA-ILD and patients with RA using the Pearson $\chi^2$ test or $t$ test,

as applicable. The correlation between patient characteristics and the different sleep variables (satisfaction, insomnia, and hypersomnia) was then analyzed using the Pearson coefficient (r).

Finally, we analyzed 6 multivariate linear regression models: 3 to identify factors associated with sleep disorders in all patients with RA and 3 for each sleep disorder in patients with RA-ILD. The variables entered into the models were those that proved to be significant in the bivariate analysis and those of clinical interest. With a significance level (alpha risk) of 0.05 and a power level (beta risk) of 0.2 in a two-sided test, the sample size calculation indicated that 30 patients were required to identify poor sleep quality in patients with RA-ILD [14]. Statistical significance was set at $p < 0.05$. The statistical analyses were performed using IBM SPSS Statistics for Macintosh, Version 28.0 (IBM Corp., Armonk, NY, USA).

## 5. Conclusions

In conclusion, patients with RA-ILD were less satisfied with their sleep and reported a greater frequency of insomnia and hypersomnia than controls. Their results were also poorer for most psychosocial factors and health-related quality of life. All these adverse events were mainly associated with the characteristics of lung disease, such as duration of ILD, inflammatory activity, and radiological pattern as well as other factors, such as age, comorbid conditions, and positive psychosocial factors. A multidisciplinary approach to detect and manage sleep disorders in RA-ILD could satisfy unmet needs in affected patients. Furthermore, it would be interesting for future studies to explore how resilience and emotional recovery affect sleep patterns in patients with RA and patients with RA-ILD and to determine how interventions to improve psychological factors could have a positive impact on the sleep quality of affected individuals.

**Supplementary Materials:** The following supporting information can be downloaded at: https://www.mdpi.com/article/10.3390/clockssleep5040049/s1, Table S1: Correlation between sleep quality and clinical characteristics of patients with RA; Table S2: Univariate and multivariate analyses of the characteristics associated with sleep disorders in RA including treatment; Table S3: Classification of the degrees of emotional intelligence according to the 24-item Trait Meta-Mood Scale (TMMS-24).

**Author Contributions:** N.M.-V. and A.F.-N. wrote the first draft of the manuscript. N.M.-V., R.R.-R., P.C.-G., A.G.-S., F.O.-M., P.B.-G., M.M.-V., I.U.-G. and S.M.-A. carried out patient recruitment and data collection. N.M.-V., R.R.-R., P.C.-G., L.C.-G. and A.F.-N. were major contributors in analysis and interpretation of data. N.M.-V. and A.F.-N. were contributors in the design of the study and interpreting the patient data and major contributor in writing the manuscript. All authors have read and agreed to the published version of the manuscript.

**Funding:** This study was funded by the FAR (Fundación Andaluza de Reumatología) (Code_2021). "Redes de Investigación Cooperativa Orientadas a Resultados en Salud (RICORS), Red de Enfermedades Inflamatorias (REI) (RD21/0002/0037)": fondos de Next Generation EU, que financian las actuaciones del Mecanismo para la Recuperación y la 4 Resiliencia (MRR).

**Institutional Review Board Statement:** The study was approved by the Research Ethics Committee of HRUM (2627-N-21). All patients gave their written informed consent before participating.

**Informed Consent Statement:** Informed consent was obtained from all subjects involved in the study.

**Data Availability Statement:** The datasets used and/or analyzed in the present study are available from the corresponding author upon reasonable request.

**Acknowledgments:** FERBT2023 The authors thank the Spanish Foundation of Rheumatology for providing medical writing/editorial assistance during the preparation of the manuscript.

**Conflicts of Interest:** The authors declare no conflict of interest.

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
