# Peer review of "Interstitial Lung Disease Is Associated with Sleep Disorders in Rheumatoid Arthritis Patients"

_2624-5175, doi:10.3390/clockssleep5040049_

Round 1

Reviewer 1 Report

Comments and Suggestions for Authors

The prevalent extra-articular expression of rheumatoid arthritis (RA) is associated with lung complications, impacting approximately 60% of RA patients over the course of the disease. RA can clinically influence various lung compartments, presenting as interstitial lung disease (ILD) or the formation of rheumatoid nodules. This can lead to pleural inflammation, effusions, and involvement of the pulmonary vasculature. Lung complications, especially RA-ILD, are linked to considerable morbidity and mortality. While lung issues in RA generally arise subsequent to joint symptoms, there are instances where pulmonary manifestations precede the onset of joint-related symptoms. If ILD is accompanied in RA patients, further discussion should be given to the degree of sleep deterioration.

Author Response

Comments for the reviewers

We would like to thank the editor for considering our work for publication in “Clocks & Sleep” and the reviewers for their comments, which have helped to improve the quality of our manuscript.

Below, we provide a point-by-point reply to the comments.

Reviewer #1: The prevalent extra-articular expression of rheumatoid arthritis (RA) is associated with lung complications, impacting approximately 60% of RA patients over the course of the disease. RA can clinically influence various lung compartments, presenting as interstitial lung disease (ILD) or the formation of rheumatoid nodules. This can lead to pleural inflammation, effusions, and involvement of the pulmonary vasculature. Lung complications, especially RA-ILD, are linked to considerable morbidity and mortality. While lung issues in RA generally arise subsequent to joint symptoms, there are instances where pulmonary manifestations precede the onset of joint-related symptoms. If ILD is accompanied in RA patients, further discussion should be given to the degree of sleep deterioration.

Reply: We agree with the reviewer and appreciate his comment and clarification. This has been included in the introduction and discussion of the document.

-Page 2; lines 141-145: “The prevalent extra-articular expression of rheumatoid arthritis (RA) is associated with lung complications, impacting approximately 60% of RA patients over the course of the disease. (4,5), leading to high morbidity and mortality associated with lung damage, greater risk of infection, severity of arthritis, and co-occurrence with other conditions (6).”

Reviewer 2 Report

Comments and Suggestions for Authors

Thank you for the opportunity to review the manuscript titled "Interstitial lung disease is associated with sleep disorders in rheumatoid arthritis patients".

1. Although this is an interesting topic, it is relatively well-recognized that patients with interstitial lung disease have poor sleep quality. Therefore, it makes sense that patients with RA-ILD have poorer sleep quality than those with RA alone. The rationale for this study should be more clearly outlined in the Introduction.

2. The number of included patients is small. If this was because the number of ILD patients was not large in your cohort, you may consider increasing the number of the matched controls.

3. Please consider using odds ratios rather than using beta coefficients.

4. Since sleep apnea is common in ILD patients, poor sleep quality and low sleep satisfaction may be associated with the presence of apnea. It has been reported that more than 50% of patients with ILD have comorbid sleep apnea.

5. Would medications have a role in causing insomnia? The medications were significantly different between the RA-ILD group and the control. Please discuss this.

6. You should consider not only the presence of ILD, but also the severity of the ILD.

7. Non-disease-related factors can also influence sleep disorders such as shift work, types of occupations, socioeconomic levels, lifestyles, alcohol drinking, substance use, sleep medication use, and etc.

8. What would be the physiologic mechanism for insomnia? It should be discussed in detail.

Author Response

Comments for the reviewers

We would like to thank the editor for considering our work for publication in “Clocks & Sleep” and the reviewers for their comments, which have helped to improve the quality of our manuscript.

Below, we provide a point-by-point reply to the comments.

Reviewer #2: Thank you for the opportunity to review the manuscript titled "Interstitial lung disease is associated with sleep disorders in rheumatoid arthritis patients".

  1. Although this is an interesting topic, it is relatively well-recognized that patients with interstitial lung disease have poor sleep quality. Therefore, it makes sense that patients with RA-ILD have poorer sleep quality than those with RA alone. The rationale for this study should be more clearly outlined in the Introduction.

Reply:  While poor sleep quality is prevalent in patients with all-cause interstitial lung disease, it may be even more pronounced in patients with rheumatoid arthritis-associated ILD (RA-ILD) due to the influence of exacerbating factors such as systemic inflammation, comorbidities, treatment, or disability associated with RA. Therefore, the objectives of this study were: (1) to explore sleep disorders in patients with RA-ILD, and (2) to identify factors associated with sleep disorders. Our research aims to enhance the understanding of the specific sleep needs of RA-ILD patients and to formulate targeted interventions for improving the quality of life for those affected.

-Page 2; lines 177-185: “While poor sleep quality is prevalent in patients with all-cause interstitial lung disease, it may be even more pronounced in patients with RA-ILD due to the influence of exacerbating factors such as systemic inflammation, comorbidities, treatment, or disability associated with RA. Therefore, the objectives of the present study were as follows: (1) to explore sleep disorders in patients with RA-ILD; and (2) to identify factors associated with sleep disorders. Our study will help to better understand the needs of the patient with RA-ILD in terms of sleep and to design specific interventions aimed at improving the quality of life of affected patients.”

  1. The number of included patients is small. If this was because the number of ILD patients was not large in your cohort, you may consider increasing the number of the matched controls.

Reply: We appreciate the reviewer's comment. However, in determining the sample size, we referenced the study by Jin-Gun Cho et al. In that study, considering an alpha risk of 0.05 and a beta risk of 0.2 in a two-sided test, it was estimated that 30 subjects were required in the observed group to detect a 66% prevalence of poor sleep quality in patients with interstitial lung disease. Additionally, with a total of 70 patients included in our study (comprising 35 with RA-ILD and 35 with RA without ILD), we successfully identified variations in sleep quality among our participants. Nevertheless, in response to the reviewer's suggestions, we have acknowledged this as one of the limitations of our study. We have also incorporated data from the sample size calculation.

-Page 13; lines 523-528: “With a significance level (alpha risk) of 0.05 and a power level (beta risk) of 0.2 in a two-sided test, the sample size calculation indicated that 30 patients were required to identify poor sleep quality in patients with RA-ILD (14).”

-Page 11; lines 401-404: “On the other hand, our sample size might have been too small to detect more substantial differences; nevertheless, we did observe variances in the sleep quality between patients with RA-ILD and those with RA without ILD.”

  1. Please consider using odds ratios rather than using beta coefficients.

Reply: We appreciate the reviewer's suggestion. However, the variables related to sleep quality are quantitative, so we have used beta coefficients. Sleep quality is assessed using the Oviedo Sleep Questionnaire (OSQ), which comprises multiple items and subscales, each with numerical scores. For example, subjective satisfaction with sleep is measured on a scale from 1 to 7, where a higher score indicates greater satisfaction. Insomnia and hypersomnia are assessed with scores ranging from 9 to 45 and 3 to 15, respectively, with higher scores indicating greater severity. Therefore, the variables are quantitative in nature as they involve numerical measurements and scales for assessing different aspects of sleep quality.

  1. Since sleep apnea is common in ILD patients, poor sleep quality and low sleep satisfaction may be associated with the presence of apnea. It has been reported that more than 50% of patients with ILD have comorbid sleep apnea.

Reply: Thank you for your comment. A total of 7 patients had sleep apnea, 4 patients with RA-ILD and 3 patients with RA without ILD. This data has been added to the document.

-Page 13; line 503: “The diagnosis of sleep apnea was also included.”

Table 1: Baseline characteristics of the study population

VARIABLE

RA-ILD

N = 35

RA without ILD

N = 35

p-value

Clinical characteristics

Age, years, mean (SD)

69.7 (9.3)

66.6 (7.0)

0.130

Age >60 years, n (%)

27 (77.1)

29 (82.9)

0.550

Male sex; n (%)

20 (57.1)

20 (57.1)

1.000

Duration of RA, months, median (IQR)

149.8 (93.3-245.5)

133.7 (67.8-204.2)

0.384

Duration of ILD, months, mean (SD)

66.1 (47.2)

-

-

RF+ (>10), n (%)

33 (94.3)

31 (88.6)

0.393

High RF (>60), n (%)

24 (68.6)

17 (48.6)

0.089

ACPA+ (>20), n (%)

32 (91.4)

31 (88.6)

0.690

Radiographic erosions, n (%)

21 (60.0)

19 (55.6)

0.705

Comorbid conditions

Smoking history

0.760

    Nonsmokers, n (%)

17 (48.6)

18 (51.4)

    Exsmokers, n (%)

10 (28.6)

8 (22.9)

    Active smokers, n (%)

8 (22.9)

9 (25.7)

Dyslipidemia, n (%)

13 (37.1)

12 (53.3)

0.873

Sleep apnea, n (%)

4 (11.4)

3 (8.6)

0.690

Arterial hypertension, n (%)

19 (54.3)

13 (37.1)

0.150

Obesity (BMI>30), n (%)

6 (17.1)

6 (17.1)

1.000

Diabetes mellitus, n (%)

7 (20.0)

6 (17.1)

0.759

CCI, median (IQR)

2.0 (1.0-3.0)

1.0 (1.0-2.0)

0.099

Age-CCI, median (IQR)

4.0 (3.0-5.0)

3.0 (3.0-5.0)

0.042

Treatment

   csDMARDs, n (%)

28 (80.0)

33 (94.3)

0.074

      Methotrexate, n (%)

19 (54.3)

27 (77.1)

0.040

      Leflunomide, n (%)

3 (8.6)

6 (17.1)

0.284

      Sulfasalazine, n (%)

2 (5.7)

2 (5.7)

1.000

      Hydroxychloroquine, n (%)

6 (17.1)

0 (0.0)

0.010

      Mycophenolate, n (%)

4 (11.4)

0 (0.0)

0.032

   bDMARDs, n (%)

22 (62.9)

18 (51.4)

0.334

      Anti-TNF, n (%)

4 (11.4)

11 (31.4)

0.041

      Tocilizumab, n (%)

3 (8.6)

2 (5.7)

0.643

      Abatacept, n (%)

13 (37.1)

3 (8.6)

0.004

      Rituximab, n (%)

2 (5.7)

0 (0.0)

0.151

   JAK inhibitors, n (%)

0 (0.0)

2 (5.7)

0.151

   Glucocorticoids, n (%)

22 (62.9)

6 (17.1)

0.001

Pulmonary function tests

FVC < 80%, n (%)

28 (80.0)

5 (14.3)

<0.001

   FVC predicted (%), mean (SD)

63.0 (17.1)

83.4 (4.4)

<0.001

FEV1 < 80%, n (%)

23 (67.6)

5 (14.3)

0.001

   FEV1 predicted (%), mean (SD)

68.7 (15.9)

84.0 (11.5)

<0.001

DLCO <80%, n (%)

29 (85.3)

3 (8.6)

<0.001

   DLCO-SB predicted (%), mean (SD)

61.0 (15.2)

85.9 (7.9)

<0.001

HRCT pattern

   UIP, n (%)

29 (82.9)

0 (0.0)

<0.001

   NSIP, n (%)

6 (17.1)

0 (0.0)

<0.001

Abbreviations. RA: rheumatoid arthritis; ILD: interstitial lung disease; SD: standard deviation; RF: rheumatoid factor; ACPA: anti–citrullinated peptide antibodies; CCI: Charlson Comorbidity Index (CCI); Age-CCI: age-adjusted CCI; csDMARDs: conventional synthetic disease-modifying antirheumatic drugs; bDMARDs: biological disease-modifying antirheumatic drugs; FVC: forced vital capacity; FEV1: forced expiratory volume in 1 second; DLCO: diffusing capacity of the lungs for carbon monoxide; HRCT: high-resolution computed tomography; UIP: usual interstitial pneumonia; NSIP: nonspecific interstitial pneumonia.

  1. Would medications have a role in causing insomnia? The medications were significantly different between the RA-ILD group and the control. Please discuss this.

Reply: In recent years, there has been a growing body of evidence supporting the efficacy and safety of certain medications in this patient population. Specifically, patients with RA-ILD are often recommended to receive treatment regimens that include glucocorticoids, hydroxychloroquine, mycophenolate, and abatacept. The rationale behind this approach stems from the positive outcomes observed in studies and clinical practice guidelines (Rodríguez Portal JA, et al. SER-SEPAR recommendations for the management of rheumatoid arthritis-related interstitial lung disease. Part 1: Epidemiology, risk factors and prognosis. Reumatol Clin (Engl Ed). 2022 Oct;18(8):443-452. doi: 10.1016/j.reumae.2022.02.004; Narváez J, et al. SER-SEPAR recommendations for the management of rheumatoid arthritis-related interstitial lung disease. Part 2: Treatment. Reumatol Clin (Engl Ed). 2022 Nov;18(9):501-512. doi: 10.1016/j.reumae.2022.03.004.), highlighting the effectiveness of these medications in managing both the rheumatoid arthritis and the associated interstitial lung disease.

Moreover, in response to the reviewer's suggestion, we conducted a comprehensive multivariate analysis, incorporating variables such as methotrexate, biological drugs, mycophenolate, and glucocorticoids. Notably, a similar model emerged from this analysis, reinforcing the robustness of our findings. We have incorporated this data into the document.

-Page 7; lines 285-287: “We conducted an alternative multivariate model, adjusting for treatment with methotrexate, biological therapy, mycophenolate, and glucocorticoids, with comparable results (Supplementary Table 2).”

Supplementary table 2: Univariate and multivariate analyses of the characteristics associated with sleep disorders in RA including treatment.

Predictor

Univariate

Multivariate

B

95% CI

B

95% CI

p-value

Sleep satisfaction*

   Female sex

0.308

-0.134, 0.751

   Age > 60 years

-0.771

-1.310, -0.231

-0.624

-1.241, -0.007

0.048

   ILD

-0.486

-0.914, -0.058

-0.400

-0.894, -0.032

0.046

   Age-CCI

-0.162

-0.288, -0.035

   DAS28-ESR

-0.326

-0.544, -0.109

-0.233

-0.456, -0.016

0.040

   Resilience

0.007

-0.001, 0.015

   ER

0.034

0.007, 0.062

   Depression

-0.077

-0.127, -0.027

   Methotrexate

0.138

-0.329, 0.604

   bDMARDs

0.394

-0.138, 0.926

   Mycophenolate

0.246

-1.623, 2.116

   Glucocorticoids

0.061

-0.118, 0.241

Severity of insomnia**

   Female sex

0.392

-3.101, 3.884

   Age > 60 years

-0.099

-4.544, 4.347

   ILD

6.257

3.148, 9.366

3.953

1.056, 6.851

0.008

   Age-CCI

1.240

0.291, 2.430

   DAS28-ESR

2.494

0.797, 4.192

   Resilience

-0.130

-0.184, -0.076

-0.108

-0.160, -0.056

<0.001

   ER

-0.138

-0.357, 0.081

   Depression

0.721

0.343, 1.099

   Methotrexate

1.741

-2.355, 5.836

   bDMARDs

0.041

-4.301, 4.383

   Mycophenolate

1.841

-12.951, 11.633

   Glucocorticoids

-0.534

-1.965, 0.896

Hypersomnia***

   Female sex

0.417

-0.584, 1.417

   Age > 60 years

1.320

0.332, 2.308

   ILD

1.400

0.465, 2.335

1.040

0.086, 1.995

0.033

   Age-CCI

0.329

0.043, 0.615

   DAS28-ESR

0.139

-0.379, 0.656

   Resilience

-0.020

-0.037, -0.003

   ER

-0.100

-0.159, -0.042

-0.082

-0.142, -0.021

0.010

   Depression

0.103

-0.014, 0.220

   Methotrexate

-0.306

-1.396, 0.784

   bDMARDs

0.135

-1.121, 1.391

   Mycophenolate

-2.277

-6.620, 2.065

   Glucocorticoids

-0.090

-0.504, 0.324

 *Nagelkerke R2 = 0.191;  **Nagelkerke R2 = 0.369;  ***Nagelkerke R2 = 0.184

Variables included in the equation: sex, age, diagnosis of ILD, DAS28-ESR, resilience, ER, depression, age-CCI, Methotrexate, bDMARDs, Mycophenolate,    Glucocorticoids

Abbreviations. RA: rheumatoid arthritis; ILD: interstitial lung disease; Age-CCI: age-adjusted Charlson Comorbidity Index; DAS28-ESR: 28-joint Disease Activity Score with erythrocyte sedimentation rate; ER: emotional recovery; bDMARDs: biological disease-modifying antirheumatic drugs.

  1. You should consider not only the presence of ILD, but also the severity of the ILD.

Reply: Thank you for your comment. We appreciate your feedback, and we completely agree with your suggestion. In response to your comment, we have incorporated variables related to the severity of ILD into our analysis. Specifically, we have included parameters such as the radiological pattern categorized as UIP pattern and Forced Vital Capacity (FVC) values. Notably, one of the variables that has shown a significant association with impaired sleep satisfaction in RA-ILD patients is the UIP radiological pattern. Patients with RA-ILD, particularly those with the UIP pattern, have greater disease severity, an increased risk of progression, and higher mortality rates. We have included these findings in the results and discussion section of the manuscript.

-Page 11; lines 366-371: “We also found the UIP radiologic pattern to be associated with lower sleep satisfaction in the cases. Other authors report that the most fibrotic forms of ILD, especially UIP-type ILD, have been associated with poorer quality of life, greater disease progression, and mortality in affected patients (39). Moreover, this type of interstitial involvement tends to respond worse to treatment, and the course of the disease is similar to that of idiopathic pulmonary fibrosis (40).”

Table 4: Univariate and multivariate analyses of characteristics associated with sleep disorders in RA-ILD

Predictor

Univariate

Multivariate

B

95% CI

B

95% CI

p-value

Sleep satisfaction*

   Female sex

0.150

-0.496, 0.796

   Age > 60 years

-0.792

-1.503, -0.081

-0.817

-1.458, -0.177

0.014

   Age-CCI

-0.054

-0.228, 0.120

   FVC

0.019

0.001, 0.037

   UIP pattern

-1.292

-0.240, -2.343

-1.419

-2.344, -0.493

0.004

   DAS28-ESR

-0.334

-0.659, -0.009

-0.324

-0.601, -0.046

0.024

   Resilience

0.008

-0.002, 0.018

   Depression

-0.057

-0.133, 0.020

Severity of insomnia**

   Female sex

-1.583

-7.065, 3.898

   Age > 60 years

0.449

-6.043, 6.491

   Age-CCI

1.869

0.536, 3.202

1.584

0.499, 2.669

0.006

   FVC

-0.113

-0.272, 0.046

   UIP pattern

3.979

-5.659, 10.618

   DAS28-ESR

3.423

0.743, 6.103

2.436

1.321-5.194

0.041

   Resilience

-0.118

-0.197, -0.038

-0.114

-0.179, -0.048

0.001

   Depression

0.608

-0.029, 1.244

Hypersomnia***

   Female sex

0.650

-0.988, 2.288

   Age > 60 years

2.013

0.281, 3.745

2.208

0.423-3.994

0.017

   Age-CCI

0.452

0.035, 0.869

   FVC

0.007

-0.042, 0.057

   UIP pattern

2.000

-0.837, 4.837

   DAS28-ESR

-0.342

-1.216, 0.533

   Resilience

-0.004

-0.031, 0.023

   Depression

-0.038

-0.234, 0.169

  *Nagelkerke R2 = 0.353; **Nagelkerke R2 = 0.464; ***Nagelkerke R2 = 0.136

Variables included in the equation: sex, age, age-CCI, UIP radiologic pattern, DAS28-ESR, resilience, depression.

Abbreviations. RA: rheumatoid arthritis; ILD: interstitial lung disease; Age-CCI: age-adjusted Charlson Comorbidity Index; FVC: forced vital capacity; UIP: usual interstitial pneumonia; HAQ: Health Assessment Questionnaire; DAS28-ESR: 28-joint Disease Activity Score with erythrocyte sedimentation rate.

  1. Non-disease-related factors can also influence sleep disorders such as shift work, types of occupations, socioeconomic levels, lifestyles, alcohol drinking, substance use, sleep medication use, and etc.

Reply: We acknowledge the reviewer's insight regarding factors such as occupation types, diet, stress, and drug use that may influence patients' sleep disorders. Despite these acknowledged influences, our study successfully identified disease characteristics, such as the duration of ILD, inflammatory activity, and radiological pattern, that exhibited associations with sleep disorders in RA-ILD patients. We have incorporated this suggestion in the limitations section of our study, recognizing the importance of these external influences on the broader context of our document.

-Page 11; lines 397-401: “We must also take into account the role of factors that may impact sleep disorders, such as occupation types, diet, stress, and drug use. Nevertheless, despite these considerations, we were able to demonstrate disease characteristics, including the duration of ILD, inflammatory activity, and radiological pattern, which were associated with sleep disorders in patients with RA-ILD.”

  1. What would be the physiologic mechanism for insomnia? It should be discussed in detail.

Reply:   We thank the reviewer for his comment because it allowed us to modify a phrase in the document that was poorly expressed in the discussion. In our study, patients with RA-ILD were more severe in all sleep dimensions (satisfaction, insomnia, and hypersomnia), and insomnia was more prevalent in RA-ILD patients than in patients with RA but not ILD. The elevated prevalence of comorbid conditions, heightened joint disease severity as indicated by DAS28-ESR, inflammation, and increased resilience impairment observed in patients with RA-ILD suggest potential links to the pathogenic mechanisms of insomnia. These factors collectively may contribute to the heightened prevalence of insomnia in this population, pointing towards a complex interplay between RA-related factors and sleep disturbances.

-Page 10; lines 329-331: “In accordance with these methods, findings for patients with RA-ILD were more severe in all sleep dimensions (satisfaction, insomnia, and hypersomnia), and insomnia was more prevalent in RA-ILD patients than in patients with RA but not ILD.”

-Page 10; lines 339-342: “The elevated prevalence of comorbid conditions heightened joint disease severity as indicated by DAS28-ESR, inflammation, and increased resilience impairment observed in patients with RA-ILD suggest potential links to the pathogenic mechanisms of insomnia.”

Reviewer 3 Report

Comments and Suggestions for Authors

Dear Editor

This is a very well-oriented observational, case-control study based on a prospective cohort of 35 patients with RA-ILD (cases) and compared them with a group of patients without 35 RA-ILD (controls). The case group comprised patients with RA-ILD selected consecutively between 2021 and 2022 from the RA-ILD cohort of HRUM, which has been prospectively followed up since 2015.

The study followed the right way of collecting data and it followed the appropriate statistical analysis.

The authors evaluated by using the Oviedo Sleep Questionnaire (OSQ), - a semi-structured interview comprising 15 items - the Sleep quality, insomnia, and hypersomnia of 35 patients with RA-ILD.

The unique value of this study is the way that describes sleep disorders and their association with other psychosocial factors in patients with RA with and without ILD using a wide platform of questionnaires; the Wagnild and Young Resilience Scale Trait Meta-Mood Scale (TMMS-24), Hospital Anxiety and Depression Scale (HADS), 36-item Short Form Survey (SF-36), Health 172 Assessment Questionnaire (HAQ), Functional Assessment of Chronic Illness Therapy – Fatigue scale (FACIT), Charlson 194 Comorbidity Index (CCI).

The authors concluded, in accordance with the above methods, that patients with RA-ILD were more severe in all sleep dimensions (satisfaction, insomnia, and hypersomnia) than in patients with RA but not ILD, adding to the literature a well-documented manuscript.

Author Response

Comments for the reviewers

We would like to thank the editor for considering our work for publication in “Clocks & Sleep” and the reviewers for their comments, which have helped to improve the quality of our manuscript.

Below, we provide a point-by-point reply to the comments.

Reviewer #3: Dear Editor, This is a very well-oriented observational, case-control study based on a prospective cohort of 35 patients with RA-ILD (cases) and compared them with a group of patients without 35 RA-ILD (controls). The case group comprised patients with RA-ILD selected consecutively between 2021 and 2022 from the RA-ILD cohort of HRUM, which has been prospectively followed up since 2015.

The study followed the right way of collecting data and it followed the appropriate statistical analysis.

The authors evaluated by using the Oviedo Sleep Questionnaire (OSQ), - a semi-structured interview comprising 15 items - the Sleep quality, insomnia, and hypersomnia of 35 patients with RA-ILD.

The unique value of this study is the way that describes sleep disorders and their association with other psychosocial factors in patients with RA with and without ILD using a wide platform of questionnaires; the Wagnild and Young Resilience Scale Trait Meta-Mood Scale (TMMS-24), Hospital Anxiety and Depression Scale (HADS), 36-item Short Form Survey (SF-36), Health 172 Assessment Questionnaire (HAQ), Functional Assessment of Chronic Illness Therapy – Fatigue scale (FACIT), Charlson 194 Comorbidity Index (CCI).

The authors concluded, in accordance with the above methods, that patients with RA-ILD were more severe in all sleep dimensions (satisfaction, insomnia, and hypersomnia) than in patients with RA but not ILD, adding to the literature a well-documented manuscript.

Reply: We agree with the reviewer and appreciate his/her comment.

Round 2

Reviewer 2 Report

Comments and Suggestions for Authors

The authors adequately responded to the raised concerns.